# Hybrid Electromagnetic Nanomaterials Based on Polydiphenylamine-2-carboxylic Acid

**DOI:** 10.3390/polym12071568

**Published:** 2020-07-15

**Authors:** Sveta Zhiraslanovna Ozkan, Aleksandr Ivanovich Kostev, Galina Petrovna Karpacheva, Petr Aleksandrovich Chernavskii, Andrey Aleksandrovich Vasilev, Dmitriy Gennad’evich Muratov

**Affiliations:** 1A.V. Topchiev Institute of Petrochemical Synthesis, Russian Academy of Sciences, 29 Leninsky prospect, 119991 Moscow, Russia; kostev@ips.ac.ru (A.I.K.); gpk@ips.ac.ru (G.P.K.); chern5@inbox.ru (P.A.C.); vasilev@ips.ac.ru (A.A.V.); muratov@ips.ac.ru (D.G.M.); 2Department of Chemistry Lomonosov Moscow State University, 1-3 Leninskie Gory, 119991 Moscow, Russia

**Keywords:** polydiphenylamine-2-carboxylic acid, conjugated polymers, polymer–metal–carbon nanocomposites, in situ oxidative polymerization, single-walled carbon nanotubes, Fe_3_O_4_ nanoparticles, magnetic fluids, electromagnetic nanomaterials

## Abstract

Hybrid ternary nanomaterials based on conjugated polymer polydiphenylamine-2-carboxylic acid (PDPAC) (poly-*N*-phenylanthranilic acid), Fe_3_O_4_ nanoparticles and single-walled carbon nanotubes (SWCNT) were prepared for the first time. Polymer–metal–carbon Fe_3_O_4_/SWCNT/PDPAC nanocomposites were synthesized via in situ oxidative polymerization of diphenylamine-2-carboxylic acid (DPAC) by two different ways: in an acidic medium and in the interfacial process in an alkaline medium. In an alkaline medium (pH 11.4), the entire process of Fe_3_O_4_/SWCNT/PDPAC-1 synthesis was carried out in one reaction vessel without intermediate stages of product extraction and purification. In an acidic medium (pH 0.3), to prepare the Fe_3_O_4_/SWCNT/PDPAC-2 nanocomposites, prefabricated magnetite nanoparticles were deposited on the surface of obtained SWCNT/PDPAC-2. The phase composition of the nanocomposites does not depend on the synthesis reaction medium pH. The influence of the reaction medium pH on the structure, morphology, thermal, magnetic, and electrical properties of the obtained ternary nanocomposites was studied.

## 1. Introduction

Modern technology demands the creation of new generation materials with a range of required properties. In recent decades, polymer nanocomposites have become a prominent area of current research promoted by rapid developments in both polymer science and nanotechnology [1,2,3,4,5,6]. Polymer nanocomposites have unique physicochemical properties owing to synergistic performance derived from each component. Such materials are of growing interest to researchers due to their great potentials for a wide range of applications [7,8,9,10,11,12,13,14,15,16]. Polymer–metal–carbon nanocomposites containing conjugated polymers, magnetic nanoparticles and carbon nanomaterials take a special place in this class of polymer nanocomposites [17,18]. Such ternary nanocomposites are superparamagnetic [19,20,21,22] and can effectively absorb electromagnetic radiation [20,21,22,23,24,25,26,27]. Due to the combination of their electrical and magnetic properties, these electromagnetic nanocomposites can be used for medical applications (immobilization of trypsin in a nanocomposite for protein digestion) [28]; to produce sensors [29]; supercapacitors [30,31,32]; as cathode materials for rechargeable batteries and fuel cells [33,34,35]; and as anticorrosive coatings [36].

Polyaniline, polypyrrole, polythiophene, and their derivatives are examples of conjugated polymers used to synthesize ternary nanocomposites. Graphene [30,31,32], reduced graphene oxide (RGO) [25,27,29,30,36,37], and carbon nanotubes (CNT) [19,21,26,35,38,39,40] can be used as carbonaceous nanofillers. The choice of carbon nanomaterials is explained by their excellent structural and functional properties [41,42]. Ternary nanocomposites use Fe_3_O_4_ [19,21,26,27,36,38,39,40], Co_3_O_4_ [25,30,32], Fe_2_O_3_ [29,37], FeCoO [35], and CoFe_2_O_4_ [31] as magnetic nanoparticles.

There are two most common approaches to the preparation of polymer–metal–carbon nanocomposites. The first way is the in situ oxidative polymerization of aromatic amines in the presence of magnetic nanoparticles immobilized on the surface of carbon nanomaterials [19,20,23,29,39,40]. The second approach provides the in situ oxidative polymerization of aromatic amines in the presence of carbon nanomaterials followed by the deposition of magnetic nanoparticles on the polymer surface [21,22,25,38]. Depending on the synthesis way, prefabricated magnetite nanoparticles uniform in size and shape were used to synthesize the Fe_3_O_4_/MWCNT and Fe_3_O_4_/RGO nanocomposites [19,20,23]. Graphene oxide was reduced with hydrazine [20,22,25]. Multi-walled carbon nanotubes (MWCNT) were pretreated with concentrated acids to functionalize their surface [21,23]. Modified Fe_3_O_4_ nanoparticles were added to the reaction solution containing MWCNT/PANI [21].

Earlier, we obtained hybrid nanomaterials based on poly-3-amine-7-methylamine-2-methylphenazine (PAMMP), single-walled carbon nanotubes (SWCNT), and Fe_3_O_4_ nanoparticles via in situ chemical oxidative polymerization of 3-amine-7-methylamine-2-methylphenazine hydrochloride (Neutral Red) in the presence of prefabricated Fe_3_O_4_/SWCNT nanocomposites [43]. The Fe_3_O_4_/SWCNT/PAMMP nanomaterials are superparamagnetic.

In the present study, ternary nanocomposites based on polydiphenylamine-2-carboxylic acid (PDPAC), Fe_3_O_4_ nanoparticles, and SWCNT were prepared for the first time. The Fe_3_O_4_/SWCNT/PDPAC nanomaterials were synthesized under different conditions of oxidative polymerization: in the interfacial process in an alkaline medium (pH 11.4) and in a 5 M solution of sulfuric acid (pH 0.3). A comparative analysis of the structure, morphology, thermal, magnetic, and electrical properties depending on the synthesis conditions was done.

## 2. Experimental

### 2.1. Materials

Diphenylamine-2-carboxylic acid (*N*-phenylanthranilic acid) (C_13_H_11_O_2_N) (analytical grade), sulfuric acid (reagent grade), aqueous ammonia (reagent grade), and chloroform (reagent grade), as well as iron (II) sulfate (high-purity grade) and iron (III) chloride (high-purity grade) were used as received. Ammonium persulfate (analytical grade) was purified by recrystallization from distilled water. The aqueous solutions of reagents were prepared with the use of distilled water. SWCNT from Carbon Chg, Ltd. (Moscow, Russia) were produced using the electric arc discharge technique with a Ni/Y catalyst (*d* = 1.4–1.6 nm, *l* = 0.5–1.5 µm).

### 2.2. Synthesis of Fe_3_O_4_/SWCNT

The synthesis of Fe_3_O_4_ nanoparticles immobilized on the SWCNT surface was carried out via hydrolysis of iron (II) and (III) salts mixture with a molar ratio of 1:2 in a solution of ammonium hydroxide in the presence of SWCNT at 60 °C. For that, 0.86 g of FeSO_4_·7H_2_O and 2.35 g of FeCl_3_·6H_2_O were dissolved in 20 mL of distilled water (Table 1). 0.03 g of SWCNT were added to the resulting solution and heated to 60 °C, then 5 mL of ammonium hydroxide NH_4_OH were added. The resulting suspension was heated on a steam bath to 80 °C and stirred for 0.5 h. The suspension was cooled at ambient temperature under intensive stirring for 1 h. The obtained Fe_3_O_4_/SWCNT nanocomposite was filtered off, washed with distilled water until neutral reaction of the filtrate, and vacuum-dried over KOH to constant weight.

### 2.3. Preparation of Fe_3_O_4_/SWCNT/PDPAC Nanocomposites

#### 2.3.1. Preparation of Fe_3_O_4_/SWCNT/PDPAC Nanocomposites in the Interfacial Process in an Alkaline Medium

The following method was used to prepare the Fe_3_O_4_/SWCNT/PDPAC nanocomposite in the interfacial process in an alkaline medium. First, the synthesis of Fe_3_O_4_ nanoparticles of the required concentration, immobilized on the surface of SWCNT, was carried out as described in 2.2 at 55 °C. The amount of carbon nanotubes is C_SWCNT_ = 3 and 10 wt. % relative to the monomer weight. Diphenylamine-2-carboxylic acid (DPAC) solution with the required concentration (0.15 mol/L, 1 g) in a mixture of an organic solvent—chloroform (60 mL) and NH_4_OH (5 mL)—was added to the obtained aqueous alkaline suspension of the Fe_3_O_4_/SWCNT nanocomposite. The process was conducted at 55 °C with continuous intensive stirring for 0.5 h. The suspension was cooled at room temperature with vigorous stirring for 1 h. After that, the suspension was thermostated under continuous stirring at 0 °C, and the aqueous solution of ammonium persulfate (0.3 mol/L, 1.96 g) was added. The solutions of the organic and aqueous phases were mixed in one shot without gradual dosing of the reagents. The volume ratio of organic and aqueous phases was 1:1 (*V*_total_ = 120 mL). The polymerization reaction continued for 3 h under intensive stirring at 0 °C. When the synthesis was completed, the reaction mixture was precipitated in a threefold excess of 1 M H_2_SO_4_. The obtained product was filtered off and washed repeatedly with distilled water until neutral reaction of the filtrate. Since the product was in a neutral basic form, it was vacuum dried over alkali (KOH) to constant weight. The yield was 1.058 g for the nanocomposite prepared at C_FeSO4_**_⋅_**_7H2O_ = 0.86 g and C_FeCl3_**_⋅_**_6H2O_ = 2.35 g, C_SWCNT_ = 0.03 g (3 wt. % relative to the monomer weight).

Nanocomposites based on PDPAC, SWCNT, and magnetite nanoparticles obtained in an alkaline medium were marked as Fe_3_O_4_/SWCNT/PDPAC-1. PDPAC was synthesized under the same conditions [44] to compare the structure and properties of nanocomposites and polymer, and marked as PDPAC-1.

The Fe_3_O_4_/SWCNT/PDPAC-1 nanocomposites depending on synthesis conditions in alkaline medium were marked as ***F******_x_C******_y_P-1***, where ***F***—Fe_3_O_4_, ***C***—SWCNT, ***P***—PDPAC, ***x***—the iron content according to ICP-AES data, and ***y***—the SWCNT content relative to the monomer weight (Table 1).

#### 2.3.2. Preparation of Fe_3_O_4_/SWCNT/PDPAC Nanocomposites in an Acid Medium

The following method was used to prepare the Fe_3_O_4_/SWCNT/PDPAC nanocomposite in an acidic medium. First, the SWCNT/PDPAC nanocomposite was synthesized. For that, SWCNT were added to the solution of DPAC (0.1 mol/L, 0.64 g) in 5 M H_2_SO_4_ and stirred in an ultrasonic bath at room temperature for 0.5 h. The mixture was heated to 25 °C. The amount of carbon nanotubes was C_SWCNT_ = 3 and 10 wt. % relative to the monomer weight. Then, for the oxidative polymerization of DPAC in the presence of SWCNT, an aqueous solution of ammonium persulfate (0.2 mol/L, 1.368 g) in the same solvent (1/4 of total volume, *V*_total_ = 30 mL) was added dropwise to the suspension previously thermostated at 0 °C under continuous stirring. The synthesis was carried out during 3 h under intensive stirring at 0 °C. When the reaction was completed, the mixture was precipitated in 200 mL of distilled water. The obtained SWCNT/PDPAC nanocomposite was filtered off and washed repeatedly with distilled water until there was neutral reaction of the filtrate. The synthesis of Fe_3_O_4_ nanoparticles was carried out by a known method via hydrolysis of iron (II) and (III) salts mixture in a solution of ammonium hydroxide [45]. To synthesize Fe_3_O_4_ nanoparticles of the required concentration, FeSO_4_·7H_2_O and FeCl_3_·6H_2_O with a molar ratio of 1:2 were dissolved in 20 mL of distilled water and heated to 60 °C, then 5 mL of ammonium hydroxide NH_4_OH were added. The reaction mixture was heated on a steam bath to 80 °C and stirred for 0.5 h. Freshly prepared SWCNT/PDPAC nanocomposite was added immediately, without preliminary drying, to the obtained Fe_3_O_4_ suspension. The reaction mixture was stirred in an ultrasonic bath at ambient temperature for 0.5 h; then, to remove the liquid phase, it was heated to 60 °C in a crystallizing pan until precipitation. The obtained product was filtered off, washed repeatedly with distilled water to an uncolored filtrate, and to avoid dedoping, vacuum dried over CaCl_2_ to constant weight. The yield was 0.82 g for the nanocomposite prepared at C_FeSO4_**_⋅_**_7H2O_ = 0.43 g and C_FeCl3_**_⋅_**_6H2O_ = 1.175 g, with C_SWCNT_ = 0.019 g (3 wt. % relative to the monomer weight).

Nanocomposites based on PDPAC, SWCNT, and magnetite nanoparticles obtained in an acidic medium were marked as Fe_3_O_4_/SWCNT/PDPAC-2. To compare the structure and properties of nanocomposites and polymer, PDPAC was synthesized under the same conditions [46,47] and marked as PDPAC-2.

The Fe_3_O_4_/SWCNT/PDPAC-2 nanocomposites depending on synthesis conditions in acidic medium were marked as ***F******_x_C******_y_P-2***, where ***F***—Fe_3_O_4_, ***C***—SWCNT, ***P***—PDPAC, ***x***—the iron content according to ICP-AES data, and ***y***—the SWCNT content relative to the monomer weight (Table 1).

### 2.4. Preparation of Suspensions for Magnetic Fluids

The suspensions of the Fe_3_O_4_/SWCNT and Fe_3_O_4_/SWCNT/PDPAC nanocomposites in ethanol were prepared. For this, 0.01 g of the nanocomposite was added to 5 mL of ethanol, mixed thoroughly, and left in a closed vessel to prevent evaporation of the solvent.

### 2.5. Characterization

The molecular mass of diphenylamine-2-carboxylic acid polymers was measured by GPC using Water’s 150C chromatograph (San Diego, CA, USA) equipped with PLgel 5 μm MIXED-C GPC columns (Santa Clara, CA, USA), using N-methylpyrrolidone as an eluent at 60 °C. The eluent flow rate was 1 mL/min. Volume of the injected sample was 150 mL. Calibration was performed on polystyrene. RI-detector was used. Accuracy of MM definition is ~5%.

The metal content in the nanocomposites was measured quantitatively by an inductively coupled plasma atomic emission spectroscopy method (ICP-AES) using a Shimadzu ICP emission spectrometer (ICPE-9000).

FTIR spectra were measured in air on a Bruker IFS 66v FTIR spectrometer (Karlsruhe, Germany) in the range of 400–4000 cm^−1^ and analyzed with Soft-Spectra software. The samples were prepared as KBr pressed pellets. Attenuated total reflection (ATR) FTIR spectra of the samples in the attenuated total reflectance mode were recorded using a HYPERION-2000 IR microscope (Bruker, Karlsruhe, Germany) coupled with the Bruker IFS 66v FTIR spectrometer in the range of 600–4000 cm^−1^ (150 scans, ZnSe crystal, resolution of 2 cm^−1^).

An X-ray diffraction study was performed in ambient atmosphere using a Difray-401 X-ray diffractometer (Scientific Instruments Joint Stock Company, Saint-Petersburg, Russia) with Bragg–Brentano focusing on Cr*K*_α_ radiation, *λ* = 0.229 nm. The results of X-ray diffraction analysis were used to calculate the size distribution of the coherent scattering regions of crystallites [48] in magnetic nanoparticles.

An electron microscopic study was performed using a LEO912 AB OMEGA transmission electron microscope (Bioz Inc., Los Altos, CA, USA) and a Hitachi TM 3030 scanning electron microscope (Hitachi High-Technologies Corporation, Fukuoka, Japan) with magnification up to 30,000 and 30 nm resolution. The size of nanoparticles is determined using the EsiVision software (eVision Software, The Hague, The Netherlands).

A vibration magnetometer was used to study the magnetic characteristics of the systems. The cell of the vibration magnetometer was designed as a flow quartz microreactor, which made it possible to study chemical transformations in the in situ mode [49]. Specific magnetization depending on the magnetic field value was measured; magnetic characteristics of the samples at room temperature were determined.

The AC conductivity was measured with a 6367A precision LCR-meter (Microtest Co., Ltd., New Taipei City, Taiwan) in the frequency range of 0.1 Hz–1.0 MHz.

Thermogravimetric analysis (TGA) was performed on a Mettler Toledo TGA/DSC1 (Giessen, Germany) in the dynamic mode in the range of 30–1000 °C in air and in the argon flow. The weight of the samples was 100 mg, the heating rate was 10 °C/min, and the argon flow velocity was 10 mL/min. Calcined aluminum oxide was used as a reference. The samples were analyzed in an Al_2_O_3_ crucible.

Differential scanning calorimetry (DSC) was performed on a Mettler Toledo DSC823^e^ calorimeter (Giessen, Germany). The samples were heated at the rate of 10 °C/min in the nitrogen atmosphere, with the nitrogen flow rate of 70 mL/min. The measurement results were processed with the service program STARe supplied with the device.

## 3. Results and Discussion

### 3.1. Synthesis and Characterization of Nanomaterials

Synthesis methods were developed to prepare polymer–metal–carbon nanocomposites based on polydiphenylamine-2-carboxylic acid (PDPAC) first synthesized by the authors [44,46], single-walled carbon nanotubes (SWCNT), and magnetite nanoparticles (Fe_3_O_4_). Hybrid Fe_3_O_4_/SWCNT/PDPAC nanomaterials were synthesized via in situ oxidative polymerization of diphenylamine-2-carboxylic acid (DPAC) in an acidic medium and in the interfacial process in an alkaline medium. Table 1 presents the conditions for synthesis of the nanocomposites. For comparison, polymers of diphenylamine-2-carboxylic acid were synthesized under the same conditions: in a NH_4_OH solution in the presence of chloroform (PDPAC-1) and in 5 M H_2_SO_4_ (PDPAC-2). According to the GPC, the molecular mass of PDPAC-1 reached *M_w_* = 2.6 × 10^4^, the degree of polymerization was more than 120, and the polydispersity index was 2.2 [44]. The PDPAC-2 molecular mass achieved *M_w_* = 1.1 × 10^4^, the polymerization degree was above 50, and the index of polydispersity was 2.0 [46]. The oxidative polymerization in the heterophase system in the presence of chloroform reduces the probability of oxidative hydrolysis and macromolecules destruction owing to separation of reaction products and the oxidizer. This makes it possible to increase the molecular mass of PDPAC-1 by a factor of nearly 2.5 as compared to the homogeneous process.

Figure 1 shows the sequence of synthesis steps of the ternary nanomaterials.

In an alkaline medium (pH 11.4), the entire process of Fe_3_O_4_/SWCNT/PDPAC-1 synthesis was carried out in one reaction vessel without intermediate stages of disengagement and purification. The synthesis of Fe_3_O_4_ nanoparticles; their immobilization on the surface of SWCNT with the formation of Fe_3_O_4_/SWCNT nanocomposite; the anchorage of a monomer to Fe_3_O_4_/SWCNT by adding a DPAC solution in a mixture of chloroform and NH_4_OH; and in situ interfacial oxidative polymerization in the presence of ammonium persulfate were performed stepwise (Figure 1a).

In an acidic medium (pH 0.3), to prepare the Fe_3_O_4_/SWCNT/PDPAC-2 nanocomposites, the in situ oxidative polymerization of DPAC in the presence of SWCNT was performed followed by the deposition of prefabricated magnetite nanoparticles on the surface of obtained SWCNT/PDPAC-2 (Figure 1b).

The formation of Fe_3_O_4_/SWCNT/PDPAC nanocomposite materials was confirmed by X-ray diffraction (XRD), FTIR spectroscopy, transmission electron microscopy (TEM), and scanning electron microscopy (SEM).

The data presented in Figure 2 allow us to conclude that the phase composition of the nanocomposites does not depend on the synthesis reaction medium pH. According to the XRD data, irrespective of the synthesis method, diffractograms of Fe_3_O_4_/SWCNT/PDPAC identify reflection peaks of Fe_3_O_4_ in the range of scattering angles 2θ = 46.1°, 54.3°, 66.8°, 84.8°, 91.2°, and 102.2° (Cr*K*_α_ radiation) [43]. These diffraction peaks correlate to Miller indices (220), (311), (400), (422), (511), and (440), respectively, and refer to the cubic structure of Fe_3_O_4_ (JCPDS 19-0629) [50]. As can be seen in Figure 3, electron diffraction confirms the crystalline nature and phase composition of Fe_3_O_4_ nanoparticles. The diffraction rings correspond to Fe_3_O_4_ crystals. Since the single plane of SWCNT does not give a diffraction pattern, there is no carbon phase reflection peak at 2θ = 39.8°. An amorphous halo in the range of scattering angles 2θ = 20°–43° in the diffraction patterns of nanocomposites characterizes the polymer component (Figure 2b). The small wide diffraction peaks in the range of 2θ = 20°–43° characterize the crystalline DPAC oligomers contained in the polymer fraction of nanomaterials obtained in the acidic medium (Figure 2c), which is also confirmed by the DSC data. Removal of the crystalline low molecular weight fraction by washing nanocomposites with acetone can lead to a dedoping of the polymer component.

Table 2 demonstrates the volume fraction of materials. The fraction of amorphous and crystalline phases was calculated from the ratio of the areas of reflection peaks of the corresponding phases using standard samples. This technique is described in detail in [51]. As can be seen, the volume fraction of the crystalline low molecular weight fraction is negligible. Volume fraction of Fe_3_O_4_ in the nanocomposites reaches 62–95%.

We have previously shown that, during the hydrolysis of iron (II) and (III) salts mixture in the solution of ammonium hydroxide in the presence of SWCNT, the synthesis of Fe_3_O_4_ nanoparticles and their immobilization on the surface of SWCNT with the formation of the metal–carbon Fe_3_O_4_/SWCNT nanocomposite occur simultaneously [43]. The XRD, TEM, and SEM data confirm the formation of Fe_3_O_4_/SWCNT (Figure 2a, Figure 4a, and Figure 5a). According to the FTIR data, during the Fe_3_O_4_/SWCNT/PDPAC-1 synthesis in the interfacial process in an alkaline medium, the polymer interacts with the Fe_3_O_4_/SWCNT surface by binding the carboxylate ion and iron to form the Fe–OOC bond. The FTIR spectra of Fe_3_O_4_/SWCNT/PDPAC-1 nanomaterials (Figure 6a) show a hypsochromic shift of the absorption band at 561 to 578 cm^−1^, which corresponds to stretching vibrations of the ν_Fe–O_ bond. The intensity of this band grows with the increase of Fe_3_O_4_ in the Fe_3_O_4_/SWCNT/PDPAC-1 nanocomposites. There is also a long-wavelength shift in the absorption band of the stretching vibrations of ν_C=O_ bonds in the carboxyl group to 1672 cm^−1^ compared to the location of this band in the polymer at 1682 cm^−1^ (Figure 7a).

It is shown that the size and shape of Fe_3_O_4_ nanoparticles do not depend on the pH of the synthesis reaction medium as well. Spherical Fe_3_O_4_ nanoparticles are clearly visible in TEM and SEM images of the Fe_3_O_4_/SWCNT/PDPAC nanocomposites (Figure 4 and Figure 5). However, the presence of an organic solvent (chloroform) in an alkaline medium leads to the formation of a polymer morphology with distinct cavities (Figure 5b). These cavities appear in places of chloroform droplets during the precipitation of the Fe_3_O_4_/SWCNT/PDPAC-1 nanocomposites in a sulfuric acid solution. According to the TEM data, the size of Fe_3_O_4_ nanoparticles in the nanomaterials is within the range of 2 < *d* < 14 nm irrespective of the obtaining method (Figure 4). The size distribution of coherent scattering regions in the Fe_3_O_4_ nanoparticles was calculated using the XRD data (Figure 8). In the Fe_3_O_4_/SWCNT/PDPAC nanocomposites, about 80–90% of Fe_3_O_4_ crystallites are up to 5 nm in size. According to the ICP-AES, depending on the synthesis conditions, the content of iron is C_Fe_ = 6.2–34.7%. In the obtained ternary hybrid nanomaterials, the problem of agglomeration of nanocomposite components is solved by the fact that the polymer structure contains functional groups that interact both with magnetite nanoparticles and with carbon nanotubes, preventing their aggregation.

Thus, regardless of the synthesis method, the Fe_3_O_4_/SWCNT/PDPAC-1 and Fe_3_O_4_/SWCNT/PDPAC-2 nanocomposites have the same phase composition: they contain PDPAC, carbon nanotubes, and Fe_3_O_4_ nanoparticles. However, the Fe_3_O_4_/SWCNT/PDPAC-1 nanomaterials prepared in the interfacial process in an alkaline medium are metal–carbon Fe_3_O_4_/SWCNT nanocomposites coated with PDPAC-1, whereas the Fe_3_O_4_/SWCNT/PDPAC-2 nanocomposites obtained in an acidic medium are SWCNT dispersed in a polymer matrix with Fe_3_O_4_ nanoparticles immobilized on the SWCNT/PDPAC-2 surface (Figure 4 and Figure 5).

Comparison of FTIR spectra of polymers and nanocomposites prepared under the same conditions (Figure 6 and Figure 7) shows that all the main bands characterizing the chemical structure of PDPAC are retained in the FTIR spectra of Fe_3_O_4_/SWCNT/PDPAC regardless of the obtaining method. As can be seen in Figure 6 and Figure 7, the chemical structure of the polymer component depends strongly on the pH of the synthesis reaction medium. It was established that during the synthesis of Fe_3_O_4_/SWCNT/PDPAC-1 nanocomposites in an alkaline medium (pH 11.4), the polymer chain grows via the C–C bonding of phenyl rings in 2- and 4-positions in relation to nitrogen (δ_C__–__H_ = 828 and 753 cm^−1^). This differs significantly from the type of C–C bonding that takes place during the synthesis of Fe_3_O_4_/SWCNT/PDPAC-2 nanocomposites in an acidic medium (pH 0.3), when the polymer chain grows via the C–C bonding of phenyl rings in the *para* position relative to nitrogen (δ_C__–__H_ = 892 and 803 cm^−1^). The chemical structure of the corresponding polymer components is shown in Figure 1. This dependence of the polymer matrix chemical structure on the reaction medium pH was observed during the synthesis of SWCNT/PDPAC nanocomposites in the absence of Fe_3_O_4_ nanoparticles [52].

As the content of carbon nanotubes increases, ATR FTIR spectra of Fe_3_O_4_/SWCNT/PDPAC-1 nanocomposites (Figure 7a) demonstrate a bathochromic shift in the PDPAC-1 skeletal vibration frequencies at 10–16 cm^−1^, which indicates the π–π^*^ interaction of aromatic units of the polymer with the Fe_3_O_4_/SWCNT surface in an alkaline medium (stacking effect). The polymer formation on the Fe_3_O_4_/CNT surface provides the charge transfer, which is manifested in the shift of skeletal oscillation frequencies of the polymer [53,54,55].

As the content of carbon nanotubes grows, the FTIR spectra of Fe_3_O_4_/SWCNT/PDPAC-2 nanocomposites show the shift of absorption bands at 1659 and 1226 cm^−1^, associated with stretching vibrations of ν_C=__O_ in COOH groups, as well as the shift of the band at 561 to 583 cm^−1^, corresponding to stretching vibrations of the ν_Fe–O_ bond (Figure 6 and Figure 7). This indicates the interaction of carboxyl groups of SWCNT/PDPAC-2 with Fe_3_O_4_.

### 3.2. Electrical Characterization of Nanomaterials

Frequency dependence on the AC conductivity of Fe_3_O_4_/SWCNT/PDPAC nanocomposites was studied in the frequency range of 0.1 Hz–1.0 MHz (Figure 9). According to [56,57,58], the frequency dependence on the conductivity (σ_ac_) is described by evaluation:σ_ac_ = σ_dc_ + *Aω^n^*,
where *ω* = 2πf is the angular frequency,*σ*_dc_—the frequency independent (dc) part of conductivity,*n*—the exponential parameter (0 ≤ n ≤ 1),*A*—the thermally activated quantity.*A* and *n* depend on the temperature and the volume fraction of the conducting component.

Table 3 lists the conductivity values of materials. It can be seen that the conductivity (σ_ac_) of nanocomposites depends on the pH of the synthesis reaction medium.

At low frequencies, the Fe_3_O_4_/SWCNT/PDPAC-1 nanocomposites as well as Fe_3_O_4_/SWCNT are characterized by weak frequency dependence on electrical conductivity. At the same time, the electrical conductivity of nanocomposites is several orders of magnitude higher than that of PDPAC-1 (3.1 × 10^−12^ S/cm) and amounts to 3.5 × 10^−9^ S/cm (F_34_C_3_) and 8.1 × 10^−9^ S/cm (F_34_C_3_P-1) at 0.1 Hz (Figure 9a). Apparently, the conductivity of Fe_3_O_4_/SWCNT/PDPAC-1 is contributed significantly by the metal–carbon Fe_3_O_4_/SWCNT nanocomposite, i.e., the increase in electrical conductivity of nanomaterials is due to the presence both of SWCNT and magnetite nanoparticles. As frequency grows, the electrical conductivity of F_34_C_3_P-1 increases gradually by two orders of magnitude to 8.7 × 10^−7^ S/cm.

The electrical conductivity of Fe_3_O_4_/SWCNT/PDPAC-2 nanocomposites at low frequencies is significantly higher than the conductivity of Fe_3_O_4_/SWCNT/PDPAC-1. This is due to the doping of the polymer matrix that occurs during the synthesis of nanocomposites in an acidic medium. As can be seen in Figure 9b, the electrical conductivity of neat PDPAC-2 does not depend much on frequency and amounts to (1.4–2.2) × 10^−5^ S/cm. The F_7_C_3_P-2 nanocomposite with a low magnetite content demonstrates an increase in its conductivity by an order of magnitude compared to PDPAC-2 up to 2.1 × 10^−4^ S/cm. Conductivity of carbon nanotubes contributes to the growth in electrical conductivity as well. This also explains the fact that conductivity (σ_ac_) does not depend on frequency at low magnetite content. However, with the increase of Fe_3_O_4_ content in the F_14_C_3_P-2 and F_34_C_3_P-2 nanocomposites, the electrical conductivity drops to 4.6 × 10^−6^ S/cm and 3.1 × 10^−7^ S/cm, respectively. An important role in reducing the conductivity of nanocomposites belongs to the electrical conductivity of Fe_3_O_4_ that has a significantly lower conductivity (~10^−10^ S/cm) compared to the doped polymer and CNT. At the same time, the frequency dependence on the electrical conductivity is observed. As frequency increases, the electrical conductivity of F_14_C_3_P-2 and F_34_C_3_P-2 rises to 3.1 × 10^−5^ S/cm and 3.6 × 10^−6^ S/cm, respectively.

As can be seen in Figure 9c, the electrical conductivity of the nanocomposites depends on carbon nanotubes concentration. With the growth in SWCNT content from 3 to 10 wt. %, the electrical conductivity of Fe_3_O_4_/SWCNT/PDPAC-1 increases by two orders of magnitude from 3.6 × 10^−10^ S/cm (F_7_C_3_P-1) to 1.2 × 10^−8^ S/cm (F_7_C_10_P-1). At high frequencies, the electrical conductivity of Fe_3_O_4_/SWCNT/PDPAC-1 nanocomposites reaches (1.2–9.6) × 10^−7^ S/cm, depending on the content of SWCNT. With the increase in SWCNT content from 3 to 10 wt. %, the conductivity of Fe_3_O_4_/SWCNT/PDPAC-2 rises from 2.1 × 10^−4^ S/cm (F_7_C_3_P-2) to 3.2 × 10^−3^ S/cm (F_7_C_10_P-2). The main contribution to the growth of electrical conductivity is made by carbon nanotubes, which is confirmed by the independence of the σ_ac_ conductivity on frequency when Fe_3_O_4_ content is low.

Thus, as can be seen in Table 3, for all nanocomposites, the exponential parameter *n* lies in the range of 0 ≤ *n* ≤ 1, which is typical for disperse systems with a hopping mechanism of charge transfer [43,56,57,58,59,60]. Since the frequency dependence on the AC conductivity of nanocomposites shows an increase with the growth in current frequency, it can be argued that the effect of tunneling is minimal. At low frequencies, the DC conductivity plays an important role, the growth of which leads to a high frequency shift of the region of sharp increase in electrical conductivity.

### 3.3. Thermal Properties of Nanomaterials

TGA and DSC methods were used to study thermal stability of the nanocomposites depending on the preparation process. Figure 10 shows TGA thermograms of Fe_3_O_4_/SWCNT/PDPAC compared to PDPAC at heating up to 1000 °C in argon flow and in air. Table 4 presents the main thermal properties of the materials. As can be seen, the weight loss curves of the obtained materials have a stepwise pattern. Weight loss at low temperatures is associated with moisture removal, which is also confirmed by the DSC data. Figure 11 shows DSC thermograms of Fe_3_O_4_/SWCNT/PDPAC. There is an endothermic peak at ~105 °C on the DSC thermograms of nanocomposites. A small endothermic peak at ~273 °C in the Fe_3_O_4_/SWCNT/PDPAC-2 is associated with the decomposition of low molecular weight oligomers. When re-heated, this peak is absent. X-ray diffraction data also confirm the content of the crystalline low molecular weight fraction (0.75–2.35%) in the nanocomposites obtained in an acidic medium (Figure 2c, Table 2).

As seen in Figure 10, both in the PDPAC-1 and PDPAC-2 polymers, the weight loss at ~170 °C is connected with the removal of COOH groups [52]. In this temperature range, the DSC thermograms of polymers have an exothermic peak associated with decomposition [44,46]. The absence of weight loss in the Fe_3_O_4_/SWCNT/PDPAC nanomaterials in this temperature range is explained by the fact that the carboxylate groups of the polymer interact with either the Fe_3_O_4_ nanoparticles or the aromatic structures of SWCNT, depending on the method of nanocomposites preparation. The DSC thermograms of nanomaterials presented in Figure 11 do not show thermal effects in this temperature range.

The resulting nanomaterials are characterized by high thermal stability that exceeds the thermal stability of PDPAC significantly. Regardless of the preparation method, the Fe_3_O_4_/SWCNT/PDPAC nanocomposites lose half of their initial weight in an inert atmosphere at temperatures above 900 °C. At 1000 °C, both in an inert atmosphere and in air, the residue is ~50% (Table 4).

### 3.4. Magnetic Properties of Nanomaterials

Magnetic properties of Fe_3_O_4_/SWCNT/PDPAC nanomaterials were studied and the values of their main magnetic parameters were measured. The dependence of the magnetization on the magnitude of the applied magnetic field at room temperature is shown in Figure 12. Table 5 lists the values of the main magnetic properties of nanocomposites.

It can be seen that, regardless of the preparation method, the Fe_3_O_4_/SWCNT/PDPAC nanocomposites are superparamagnetic; this is due to the small size and high dispersion of magnetic nanoparticles [49,61]. The squareness ratio of the hysteresis loop is *к_S_* = *M_R_*/*M_S_* = 0. The saturation magnetization *M_S_* depends on the content of magnetite nanoparticles and reaches 31.6–39.4 emu/g depending on the synthesis method of nanocomposites. An increase in the concentration of SWCNT to 10 wt. % has little effect on the main magnetic parameters of the nanomaterials (Figure 12b).

The obtained hybrid nanomaterials can be promising as active components of magnetic fluids that are stable suspensions of magnetic nanoparticles in water or an organic solvent. Suspensions based on magnetic nanocomposites in ethanol were prepared. It was found that suspensions of Fe_3_O_4_/SWCNT/PDPAC-1 in ethanol remain stable for more than 8 months, whereas the Fe_3_O_4_/SWCNT/PDPAC-2 nanocomposites as well as Fe_3_O_4_/SWCNT dispersed in ethanol start sedimenting from the first minutes (Figure 13).

This behavior of the Fe_3_O_4_/SWCNT/PDPAC-1 and Fe_3_O_4_/SWCNT/PDPAC-2 nanocomposites is associated with the different chemical structures of the polymer component. The stability of the Fe_3_O_4_/SWCNT/PDPAC-1 suspension in ethanol is explained by the presence of a positive charge in the polymer shell structure of nanocomposites, which prevents the aggregation of magnetic nanoparticles. A positive charge can be formed both due to the charge transfer during the π-π^*^ interaction of aromatic polymer units with the surface of Fe_3_O_4_/SWCNT, and due to the electronic interaction of delocalized π-electrons with the free d-orbitals of an iron atom.

## 4. Conclusions

Ternary nanomaterials based on polydiphenylamine-2-carboxylic acid (PDPAC), single-walled carbon nanotubes (SWCNT), and magnetite nanoparticles in an acidic medium and in the interfacial process in an alkaline medium were obtained for the first time. The influence of the reaction medium pH on the structure, morphology, phase composition, thermal, electrical and magnetic properties of the obtained hybrid nanocomposites was studied. It was shown that the phase composition of Fe_3_O_4_/SWCNT/PDPAC as well as the size and shape of Fe_3_O_4_ nanoparticles do not depend on the reaction medium pH. The size of spherical Fe_3_O_4_ nanoparticles is within the range of 2 < *d* < 14 nm. Regardless of the preparation method, the nanomaterials are superparamagnetic (*к_S_* = *M_R_*/*M_S_* = 0). The Fe_3_O_4_/SWCNT/PDPAC nanomaterials lose half of their initial weight in an inert atmosphere at temperatures above 900 °C. Electrical conductivity of the nanomaterials depends strongly on the synthesis method as well as on the concentration both of carbon nanotubes and of Fe_3_O_4_ nanoparticles. At the same time, at low frequencies, the electrical conductivity of Fe_3_O_4_/SWCNT/PDPAC-2 nanocomposites is significantly higher than the conductivity of Fe_3_O_4_/SWCNT/PDPAC-1 due to the polymer matrix doping in an acidic medium. It was shown that, unlike Fe_3_O_4_/SWCNT/PDPAC-2, the Fe_3_O_4_/SWCNT/PDPAC-1 nanomaterials prepared in an alkaline medium form magnetic fluids in ethanol; this is connected with differences in the chemical structure of the polymer component. Therefore, the results indicate that prepared hybrid electromagnetic nanomaterials have potential applications in the manufacture of sensors, supercapacitors, rechargeable batteries, anti-corrosion coatings, materials absorbing electromagnetic radiation, etc. The ability of powders of the prepared ternary hybrid nanocomposites to form suspensions in water and organic media allows to obtain the nanocomposite-based coatings on smooth and rough surfaces. Furthermore, these ternary nanocomposite-based suspensions can be applied for the impregnation of porous supports. These processing methods are important at fabrication of electrodes for chemical current sources and energy storage, electromagnetic shielding, sensor sensitive elements, anti-corrosion protection, etc.

## Figures and Tables

**Figure 1 polymers-12-01568-f001:**
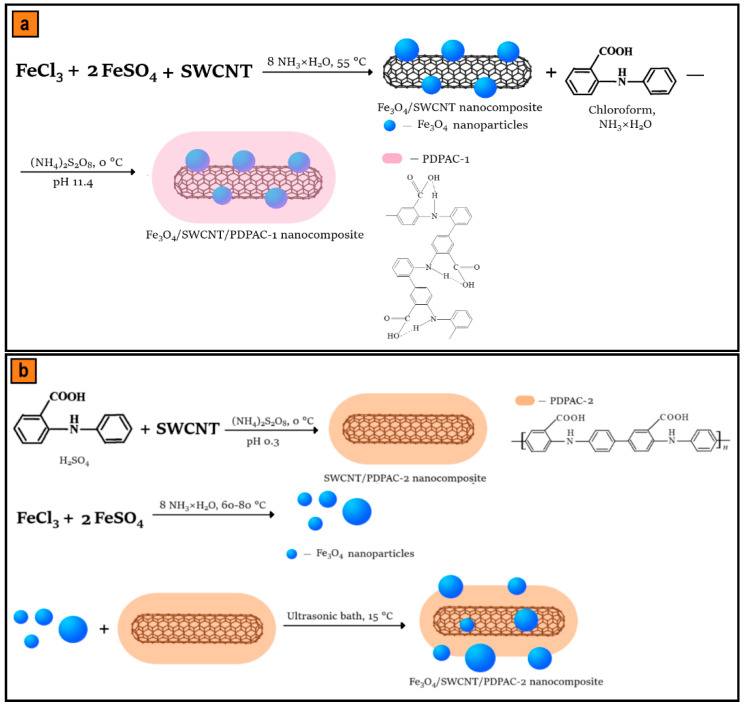
Synthesis sequence of Fe_3_O_4_/SWCNT/PDPAC in alkaline (**a**) and acidic media (**b**).

**Figure 2 polymers-12-01568-f002:**
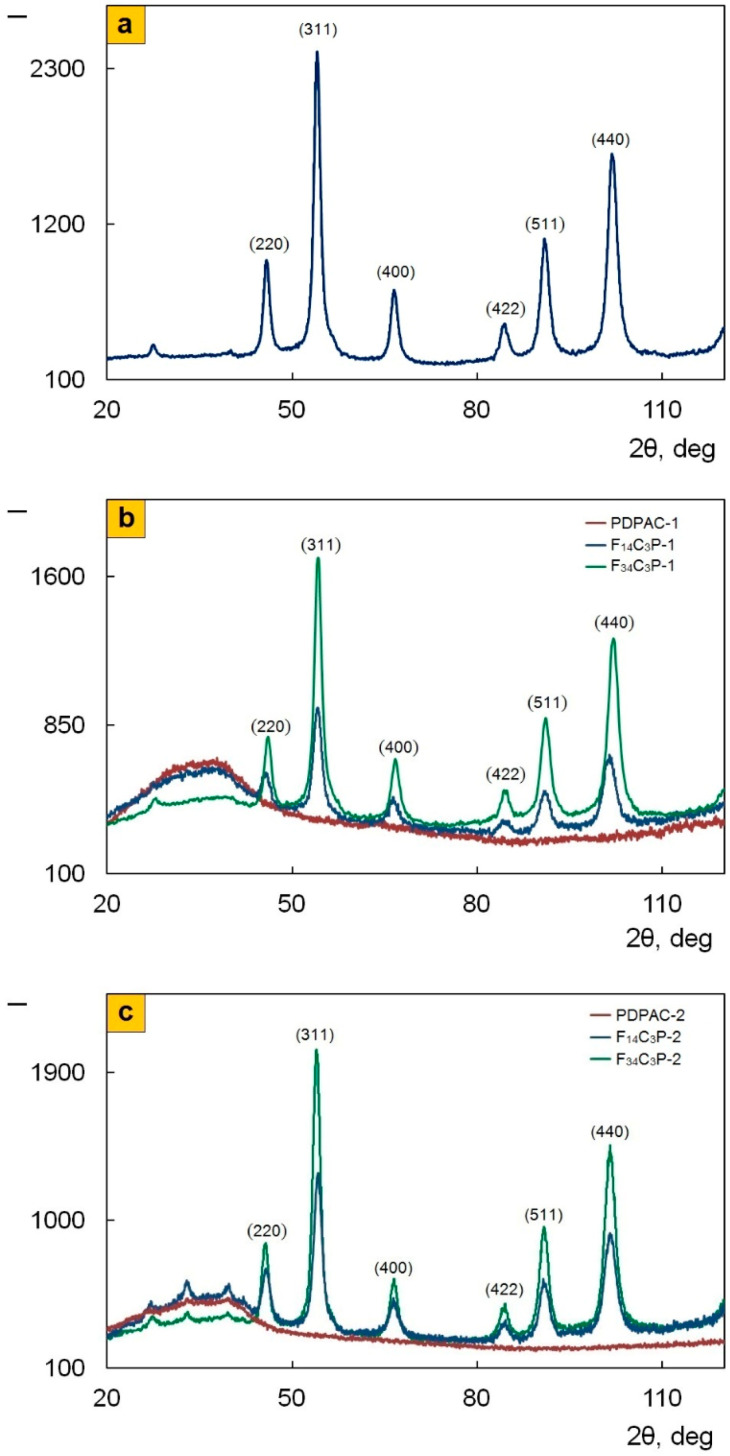
X-ray diffractograms of Fe_3_O_4_/SWCNT (**a**), PDPAC, and Fe_3_O_4_/SWCNT/PDPAC, prepared in alkaline (**b**) and acidic media (**c**).

**Figure 3 polymers-12-01568-f003:**
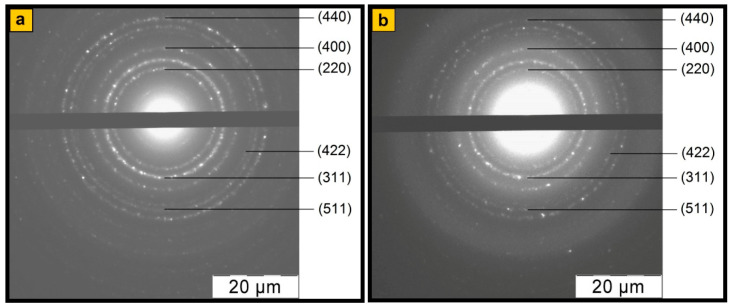
Fe_3_O_4_ diffraction images of Fe_3_O_4_/SWCNT/PDPAC, prepared in alkaline (**a**) and acidic media (**b**).

**Figure 4 polymers-12-01568-f004:**
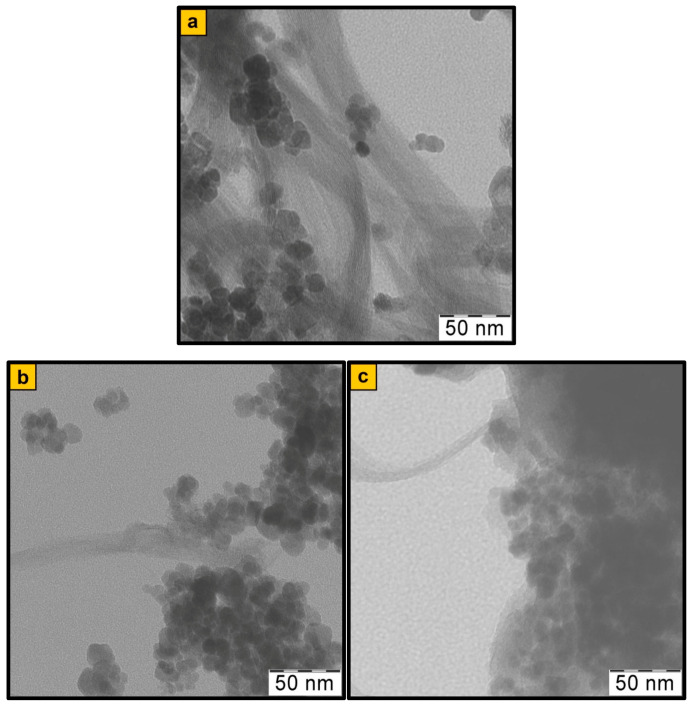
TEM images of Fe_3_O_4_/SWCNT (**a**) and Fe_3_O_4_/SWCNT/PDPAC, prepared in alkaline (**b**) and acidic media (**c**).

**Figure 5 polymers-12-01568-f005:**
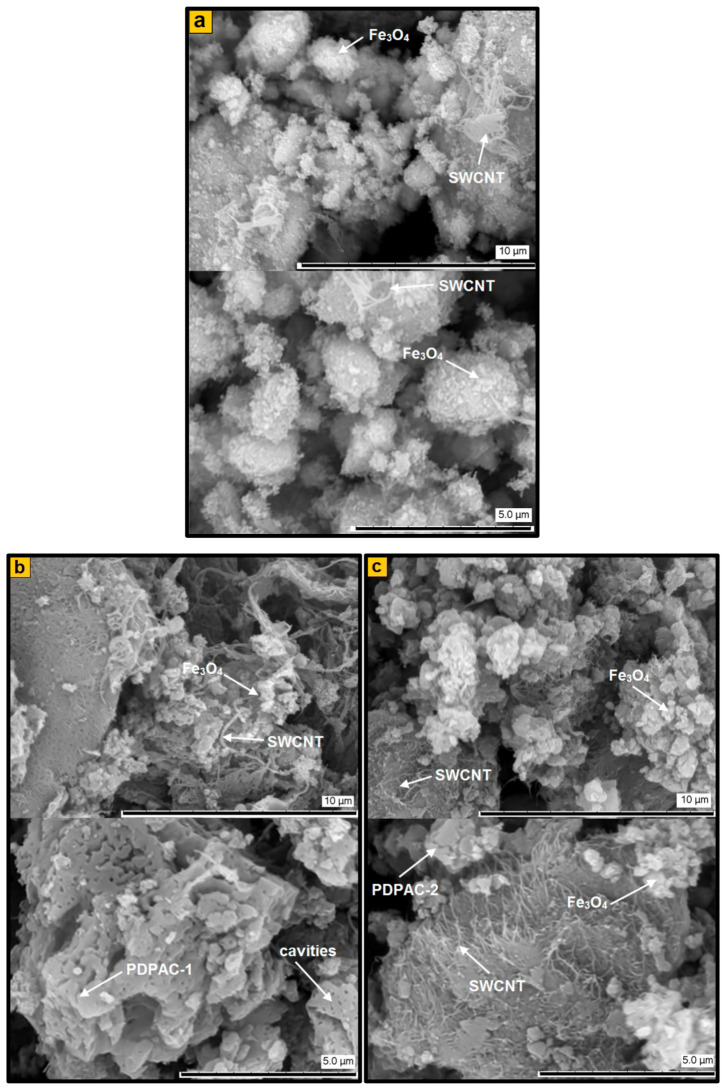
SEM images of Fe_3_O_4_/SWCNT (**a**) and Fe_3_O_4_/SWCNT/PDPAC, prepared in alkaline (**b**) and acidic media (**c**).

**Figure 6 polymers-12-01568-f006:**
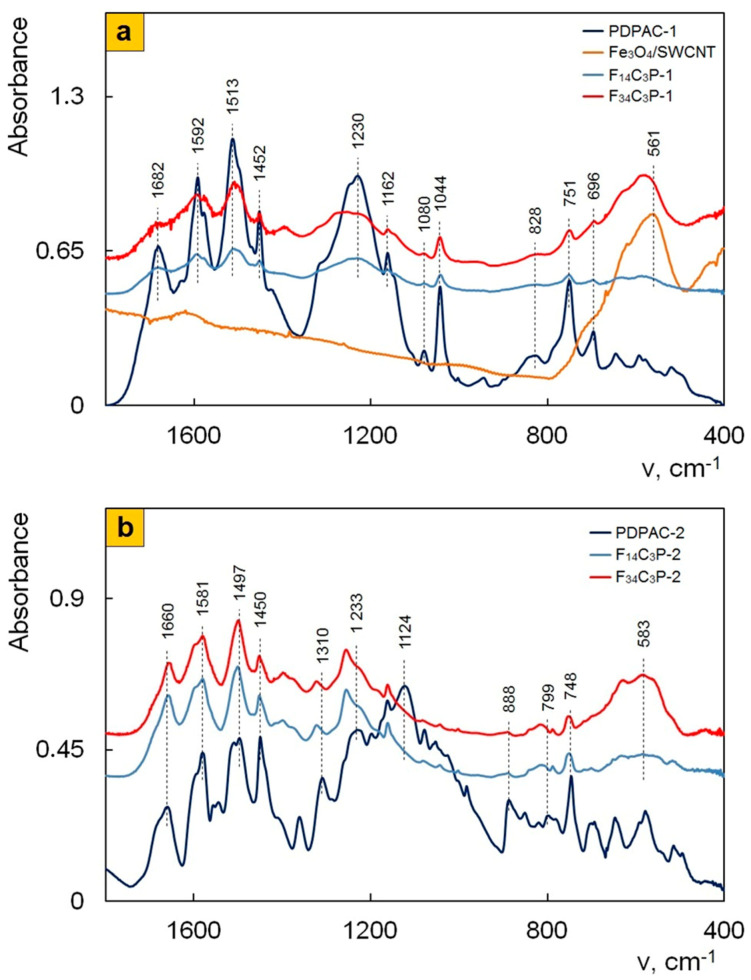
FTIR spectra of PDPAC, Fe_3_O_4_/SWCNT, and Fe_3_O_4_/SWCNT/PDPAC, prepared in alkaline (**a**) and acidic media (**b**).

**Figure 7 polymers-12-01568-f007:**
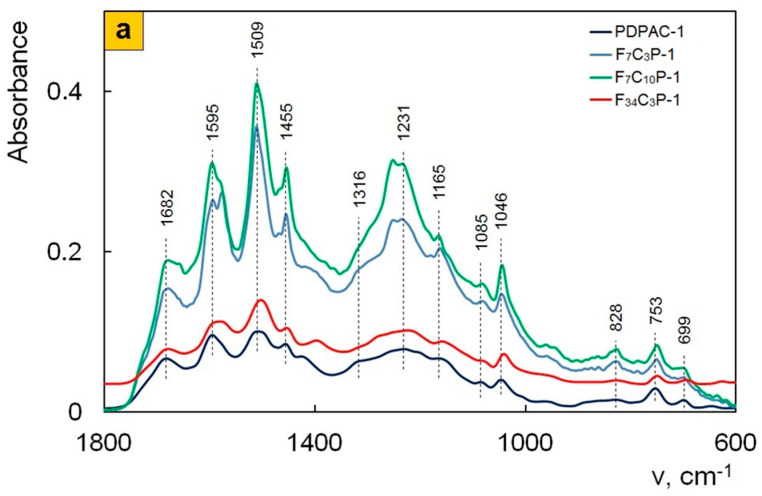
Attenuated total reflection (ATR) FTIR spectra of PDPAC and Fe_3_O_4_/SWCNT/PDPAC, prepared in alkaline (**a**) and acidic media (**b**).

**Figure 8 polymers-12-01568-f008:**
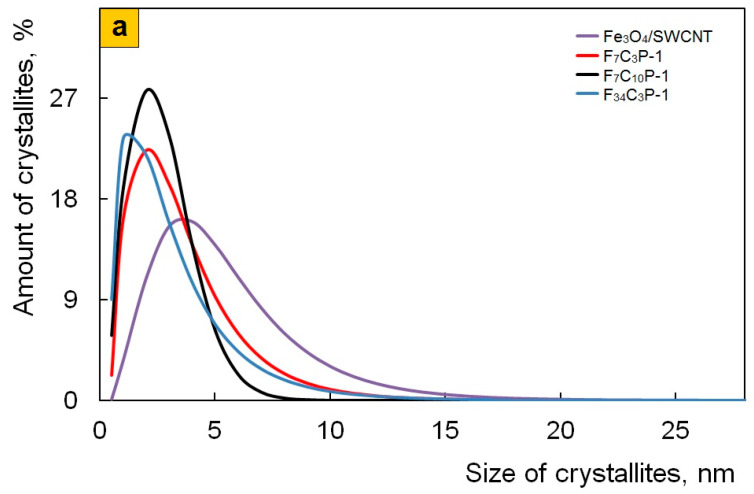
Fe_3_O_4_ crystallites size distribution in Fe_3_O_4_/SWCNT and Fe_3_O_4_/SWCNT/PDPAC, prepared in alkaline (**a**) and acidic media (**b**).

**Figure 9 polymers-12-01568-f009:**
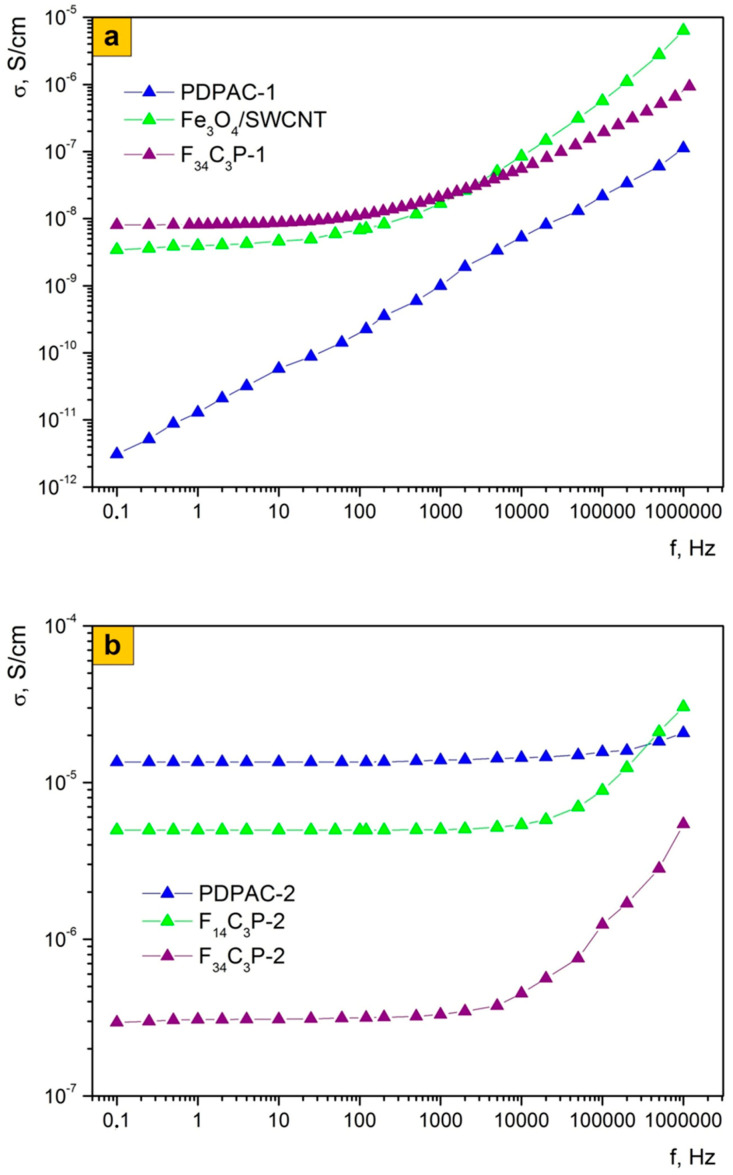
Frequency dependence of the conductivity for PDPAC, Fe_3_O_4_/SWCNT, and Fe_3_O_4_/SWCNT/PDPAC, prepared in alkaline (**a**,**c**) and acidic media (**b**,**c**).

**Figure 10 polymers-12-01568-f010:**
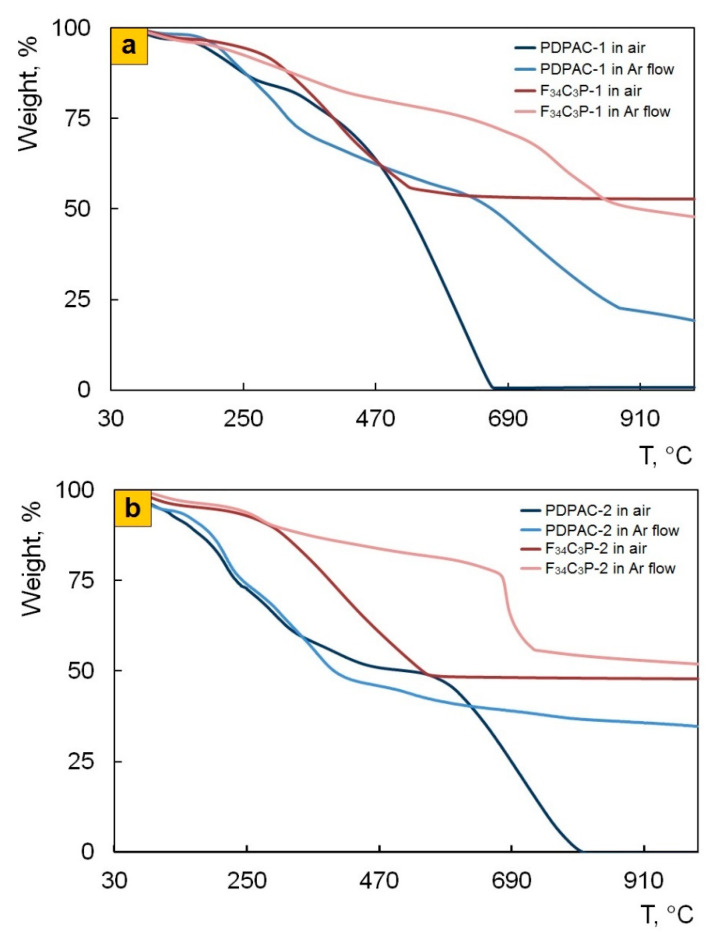
TGA thermograms of PDPAC and Fe_3_O_4_/SWCNT/PDPAC, prepared in alkaline (**a**) and acidic media (**b**), at heating up to 1000°C in the argon flow and in air.

**Figure 11 polymers-12-01568-f011:**
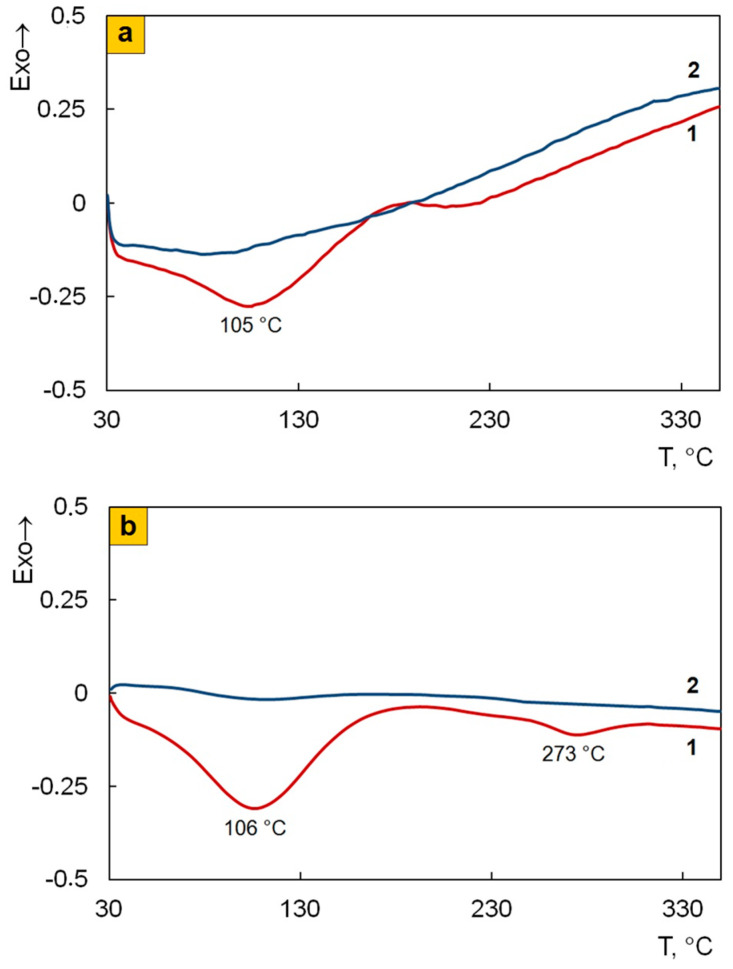
DSC thermograms of Fe_3_O_4_/SWCNT/PDPAC, prepared in alkaline (**a**) and acidic media (**b**), at heating in the nitrogen flow to 350°C (*1*—first heating, *2*—second heating).

**Figure 12 polymers-12-01568-f012:**
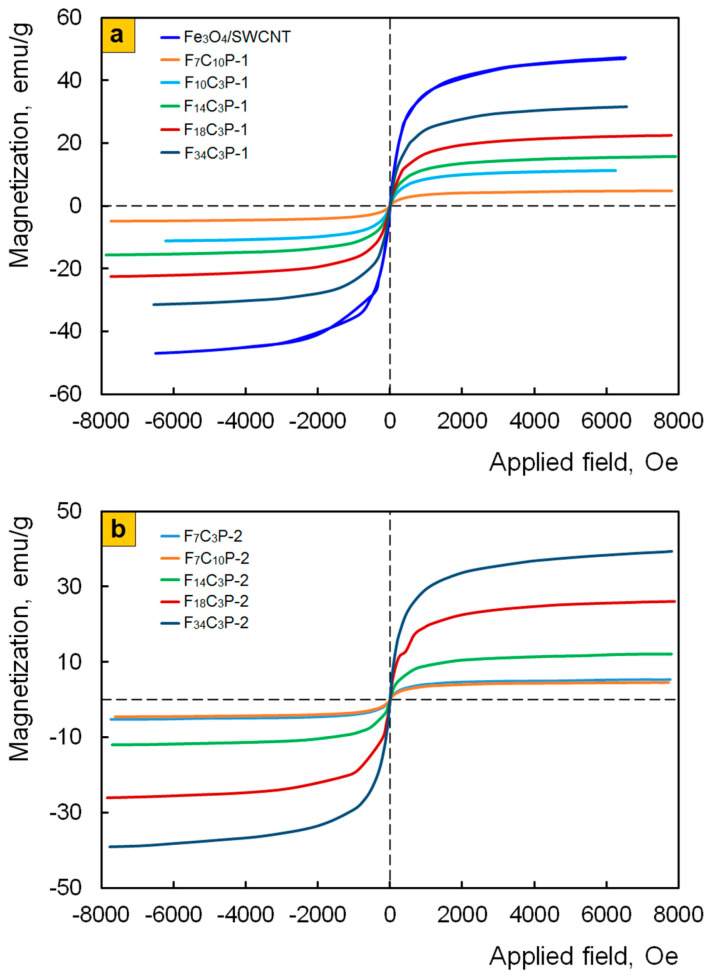
Magnetization of Fe_3_O_4_/SWCNT and Fe_3_O_4_/SWCNT/PDPAC, prepared in alkaline (**a**) and acidic media (**b**), as a function of applied magnetic field at room temperature.

**Figure 13 polymers-12-01568-f013:**
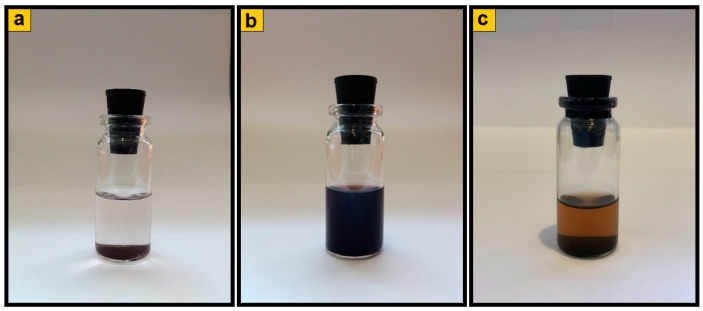
Suspensions of Fe_3_O_4_/SWCNT (**a**) and Fe_3_O_4_/SWCNT/PDPAC, prepared in alkaline (**b**) and acidic media (**c**) in ethanol. (**a**,**c**)—in 5 min, (**b**)—in 8 months.

**Table 1 polymers-12-01568-t001:** Conditions for synthesis of Fe_3_O_4_/SWCNT and Fe_3_O_4_/SWCNT/PDPAC nanocomposites.

Nanomaterials Abbreviations	Fe, % *	Preparation Method	** Amount of SWCNT, g	Amount of DPAC, g	*** Amount of Iron Salts, g	
					Fe (II)	Fe (III)
F_34_C_3_ ****	61.2	Synthesis of Fe_3_O_4_/SWCNT in an alkaline medium	0.03	0	0.86	2.35
F_34_C_3_P-1	33.5	Polymerization of DPAC in the interfacial process in an alkaline medium with Fe_3_O_4_/SWCNT	0.03	1.0	0.86	2.35
F_18_C_3_P-1	17.9				0.43	1.18
F_14_C_3_P-1	12.0				0.22	0.59
F_7_C_3_P-1	6.4				0.11	0.29
F_7_C_10_P-1	8.4		0.1			
F_34_C_3_P-2	34.7	Precipitation of Fe_3_O_4_ onto the surface of the SWCNT/PDPAC, prepared in an acidic medium	0.019	0.64	0.43	1.18
F_18_C_3_P-2	17.8				0.27	0.73
F_14_C_3_P-2	13.4				0.11	0.29
F_7_C_3_P-2	7.6				0.054	0.15
F_7_C_10_P-2	6.2		0.064			

* According to ICP-AES data. ** C_SWCNT_ = 3 and 10 wt. % relative to the monomer weight. *** FeSO_4_·7H_2_O and FeCl_3_·6H_2_O. **** Fe_3_O_4_/SWCNT.

**Table 2 polymers-12-01568-t002:** Volume fraction of materials.

Materials	Volume Fraction, %		
	Amorphous Polymer	Crystalline Polymer	Fe_3_O_4_
PDPAC-1	100.00	-	-
F_14_C_3_P-1	37.69	-	62.31
F_34_C_3_P-1	5.24	-	94.76
F_34_C_3_ *	-	-	100.00
PDPAC-2	95.60	4.40	-
F_14_C_3_P-2	28.99	2.35	68.66
F_34_C_3_P-2	6.21	0.75	93.05

* Fe_3_O_4_/SWCNT.

**Table 3 polymers-12-01568-t003:** The conductivity values of materials.

Materials	* σ_ac_, S/cm		σ_dc_, S/cm	n	A
PDPAC-1	3.1 × 10^−12^	1.1 × 10^−7^	2.8 × 10^−^^12^	0.75	8.5 × 10^−^^12^
F_7_C_3_P-1	3.6 × 10^−10^	1.2 × 10^−7^	2.4 × 10^−10^	0.82	1.7 × 10^−13^
F_7_C_10_P-1	1.2 × 10^−8^	9.6 × 10^−7^	1.1 × 10^−8^	0.96	1.9 × 10^−12^
F_34_C_3_P-1	8.1 × 10^−9^	8.7 × 10^−7^	6.8 × 10^−9^	0.99	5.4 × 10^−13^
F_34_C_3_ **	3.5 × 10^−9^	2.5 × 10^−6^	2.3 × 10^−9^	1.00	9.7 × 10^−13^
PDPAC-2	1.4 × 10^−5^	2.2 × 10^−5^	1.0 × 10^−^^5^	0.45	6.9 × 10^−^^9^
F_7_C_3_P-2	2.5 × 10^−4^	2.7 × 10^−4^	2.1 × 10^−4^	0.31	3.0 × 10^−7^
F_7_C_10_P-2	3.7 × 10^−3^	3.3 × 10^−3^	3.2 × 10^−3^	0.51	2.0 × 10^−8^
F_14_C_3_P-2	4.6 × 10^−6^	3.1 × 10^−5^	4.3 × 10^−6^	0.64	9.0 × 10^−11^
F_34_C_3_P-2	3.4 × 10^−7^	3.6 × 10^−6^	3.1 × 10^−7^	0.74	1.3 × 10^−9^

* σ—The AC conductivity at 0.1 Hz and 1.0 MHz. ** Fe_3_O_4_/SWCNT.

**Table 4 polymers-12-01568-t004:** Thermal properties of materials.

Materials	* *T*_5%_, °C	** *T*_25__%_, °C	*** *T*_50%_, °C	**** Residue, %
PDPAC-1	182/205	400/324	522/663	0/19
PDPAC-2	104/102	232/243	517/396	0/35
F_34_C_3_P-1	238/201	403/628	> 1000/910	53/48
F_34_C_3_P-2	180/225	388/677	542/ > 1000	48/52

* *T*_5%_, ** *T*_25%_, *** *T*_50%_—5, 25 and 50% weight losses (air/argon), **** residue at 1000 °C (air/argon).

**Table 5 polymers-12-01568-t005:** Magnetic properties of nanomaterials.

Nanomaterials	Fe, % *	*H_C_*, Oe	*M_S_*, emu/g	*M_R_*, emu/g	*M_R_*/*M_S_*
F_7_C_3_P-1	6.4	0	4.8	0	0
F_10_C_3_P-1	8.4	0	11.3	0	0
F_14_C_3_P-1	12.0	0	15.7	0	0
F_18_C_3_P-1	17.9	0	22.5	0	0
F_34_C_3_P-1	33.5	0	31.6	0	0
F_34_C_3_ **	61.2	6	47.3	0.45	0.009
F_7_C_3_P-2	7.6	0	5.3	0	0
F_7_C_10_P-2	6.2	0	4.5	0	0
F_14_C_3_P-2	13.4	0	12.1	0	0
F_18_C_3_P-2	17.8	0	26.1	0	0
F_34_C_3_P-2	34.7	0	39.4	0	0

* According to ICP-AES data, ** Fe_3_O_4_/SWCNT. *H_C_*—coercive force, *M_S_*—saturation magnetization, *M_R_*—residual magnetization.

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
