# Peer review of "Hybrid Electromagnetic Nanomaterials Based on Polydiphenylamine-2-carboxylic Acid"

_polymers, 2020, doi:10.3390/polym12071568_

Round 1

Reviewer 1 Report

In this manuscript, the authors prepared the hybrid ternary nanomaterials based on conjugated polymer polydiphenylamine-2-carboxylic acid (poly-N-phenylanthranilic acid), Fe3O4 nanoparticles and single-walled carbon nanotubes for the first time, and studied its electromagnetic property. The overall idea of this manuscript is clear and the logic is reasonable. It is an outstanding original work. However, there are also some questions should be addressed. I recommend this manuscript can be accepted after a mandatory revision. The detailed comments are listed as follows,

  1. Other polymer nanocomposites especially with magnetic and electric conductivity should be introduced and cited, including: https://dx.doi.org/10.30919/es8d615; https://dx.doi.org/10.30919/es8d735; Chemical modification of carbon fiber with diethylenetriaminepentaacetic acid/halloysite nanotube as a multifunctional interfacial reinforcement for silicone resin composites. Polym. Adv. Technol., 2020, 31, 527-535; https://doi.org/10.1002/tcr.202000001 ; Modification of renewable cardanol onto carbon fiber for the improved interfacial properties of advanced polymer composites. Polymers, 2020, 12, 45; Micro-crack behavior of carbon fiber reinforced Fe3O4/graphene oxide modified epoxy composites for cryogenic application. Compos. Part A, 2018, 108, 12-22; Multifunctions of polymer nanocomposites: environmental remediation, electromagnetic interference shielding, and sensing applications, ChemNanoMat, 2020, 6, 174-184; An overview of stretchable strain sensors from conductive polymer nanocomposites, J. Mater. Chem. C, 2019, 7, 11710 – 11730; Overview of carbon nanostructures and nanocomposites for electromagnetic wave shielding, Carbon, 2018, 140, 696-733.
  2. The scale bars of TEM images are not clear enough. Please enlarge the text in the SEM images.
  3. The hoping conductivity was observed in the frequency dispersion of the electrical conductivity, please explain it according to and cite the following reference: Journal of Materials Chemistry C, 2020, 8, 3029-3039.
  4. In the Abstract, I do not think it is necessary to introduce the characterization method, I advise the authors to delete these sentences. More results and conclusions of study should be mentioned in Abstract rather than introduction of characterization to support the innovation of this manuscript.
  5. I want to discuss with the authors how to solve the agglomeration of the component materials in the composites.

Reviewer 2 Report

polymers-855759

The described ternary nanomaterials are novel and have interesting properties. I recommend this article for publishing after a revision. The discussion is not extensive and sometimes simply “dry” facts are described. Moreover, I have several more specific comments:

Why the concentration of SWCNT was 3% apparently irrespectively of the amount admixed to the reaction mixture (page 3/107 and page 4/137)? Why different drying agents (KOH and CaCl2) have been used?

What is the average molecular weight of the PDPAC obtained in acidic and alkaline conditions? Experiments without Fe salts and SWCNT should give an approximate range of M and PDI.

What is indicated by the small diffraction peaks in the range corresponding to the polymer fraction (2θ 20-45o) in the samples obtained in the acidic medium (presented on Fig.2c)? Does it mean that some π-π stacking takes place in those polymers (contrary to those obtained in alkaline media)? Why you have detected π-π stacking in samples obtained in alkaline media (FTIR page 9/290) if it is not evident in X-ray diffractograms? How it corresponds with the respective frequency/conductivity dependencies (Fig.9)?

What is the amount of crystalline fraction in the composites presented in Fig. 2?

Please comment more extensively on the process of thermal decomposition. What is the reason for different behavior and what is the reason for slight differences in decomposition steps?

A broad range of applications is given (conclusions) but what would be the processing method for those nanocomposites?

Round 2

Reviewer 2 Report

The manuscript is recommended for publication in Polymers.